# Bidirectional Regulatory Effects of Warming and Winter Snow Changes on Litter Decomposition in Desert Ecosystems

**DOI:** 10.3390/plants14172741

**Published:** 2025-09-02

**Authors:** Yangyang Jia, Rong Yang, Wan Duan, Hui Wang, Zhanquan Ji, Qianqian Dong, Wenhao Qin, Wenli Cao, Wenshuo Li, Niannian Wu

**Affiliations:** 1College of Ecology and Environment, Xinjiang University, Urumqi 830046, China; 18199305490@163.com (R.Y.); 107552401736@stu.xju.edu.cn (W.D.); wanghui@stu.xju.edu.cn (H.W.); 107552301776@stu.xju.edu.cn (Z.J.); 107552301742@stu.xju.edu.cn (Q.D.); 107552201244@stu.xju.edu.cn (W.Q.); caowenli@stu.xju.edu.cn (W.C.); 19606365367@163.com (W.L.); niannianyou369@sina.com (N.W.); 2Key Laboratory of Oasis Ecology, Xinjiang University, Urumqi 830046, China

**Keywords:** desert ecosystem, warming, winter snow changes, litter decomposition, tea bag index, soil enzyme activity

## Abstract

Temperature and precipitation are the primary factors restricting litter decomposition in desert ecosystems. The desert ecosystems in Central Asia are ecologically fragile regions, and the climate shows a trend of “warm and wet” due to the regional climate change. However, the influencing mechanisms of warming and winter snow changes on litter decomposition are still poorly understood in desert ecosystems. Furthermore, the litter decomposition rate cannot be directly compared due to the large variations in litter quality across different ecosystems. Here, we simulated warming and altered winter snow changes in the field, continuously monitored litter decomposition rates of standard litter bags (i.e., red tea and green tea) and a dominant plant species (i.e., *Erodium oxyrrhynchum*) during a snow-cover and non-snow-cover period over five months. We found that warming and increased snow cover increased the litter decomposition rate of red tea, green tea, and *Erodium oxyrhinchum*, and had significant synergistic effects on litter decomposition. The effects of warming and winter snow changes on litter decomposition were more pronounced in April, when the hydrothermal conditions were the best. The decomposition rates of all three litter types belowground were higher than those on the soil surface, highlighting the important roles of soil microbes in accelerating litter decomposition. Furthermore, we found that warming and winter snow changes altered litter decomposition by influencing soil enzyme activities related to soil carbon cycling during the snow-cover period, while influencing soil enzyme activities related to soil phosphorus cycling during the non-snow-cover period. And, notably, decreased snow cover promoted soil enzyme activities during the snow-cover period. More interestingly, our results indicated that the decomposition rate (k) was the lowest, but the stability factor (S) was the highest in the Gurbantünggüt Desert based on the cross-ecosystem comparison using the “Tea Bag Index” method. Overall, our results highlighted the critical roles of warming and winter snow changes on litter decomposition. In future research, the consideration of relationships between litter decomposition and soil carbon sequestration will advance our understanding of soil carbon cycling under climate change in desert ecosystems.

## 1. Introduction

Global warming is causing notable climate change at an unprecedented rate, making it a serious challenge for the Earth [1,2]. Global warming directly leads to a rise in temperature and simultaneously causes changes in precipitation patterns [1]. Central Asia, lying in the middle of the Eurasian continent, is one of the largest arid regions at the middle latitudes and covers more than 90% temperate deserts [3,4]. Recent studies have indicated that the climate in this arid region shows a significant trend of “warm and wet”; more particularly, the winter snowfall has significantly increased [5]. Litter decomposition, a key process of nutrient recycling, links the carbon cycling between the atmosphere and soils [6,7]. Decomposition processes are primarily affected by temperature and precipitation, which have caused plant ecologists to pay more and more attention to uncover the effects of climate change on litter decomposition across different ecosystems [7,8,9,10]. Traditionally, most researchers consider increased precipitation to have stimulating effects on litter decomposition, while the impact of increasing temperature on decomposition is not always consistent due to the variation in background temperature [11,12]. However, the interactive effects of warming and winter snow changes on litter decomposition are still unclear, and rare studies reveal the potential influence patterns in desert ecosystems.

Temperature has been considered the primary factor altering litter decomposition, and warming is bound to cause significant changes in decomposition rate [6]. Soil enzyme activities are indicators of soil microbial activities and play crucial roles in the litter decomposition process, which is sensitive to global warming [13,14]. Warming directly increases soil temperature, alters soil enzyme activities, and subsequently affects litter decomposition [13,15]. Previous studies have indicated that warming increased litter decomposition rate by increasing soil β-1,4-glucosidase (BG) and alkaline phosphatase (ALP) activities [9,13]. However, prolonged high temperatures could lead to a decrease in soil enzyme activities and restrain the litter decomposition rate [16]. For example, one previous study in Moso bamboo forests found that a decrease in hydrolase activity (i.e., BG and β-1,4-N-acetylglucosaminidase (NAG)), induced by warming, reduced the litter decomposition [17]. Furthermore, warming could alter soil enzyme activities by influencing soil physicochemical properties. Previous studies have revealed that warming accelerated the mineralization of soil organic matter, increased the content of soil organic carbon and NO_3_^−^-N, and subsequently promoted soil microbial activities and litter decomposition rate in forest and grassland ecosystems [18,19]. Global meta-analysis reveals that warming poses stimulated effects on litter decomposition, which could be increased by 4.2–4.4% [11,20]. However, notably, the effects are not consistent in some field studies, which showed neutral and even negative effects on litter decomposition [9,21,22]. Previous studies found that warming significantly reduced litter decomposition rate in arctic ecosystems, especially in summer, when soil moisture was relatively low [21,22]. These results highlight that the effects of warming on litter decomposition are also strongly affected by soil moisture [22]. For desert ecosystems, the background air temperature is relatively high, and the soil environment (e.g., water and nutrients) is harsh, so warming could increase evapotranspiration and intensify the deficiency of soil moisture [23]. The influence patterns of warming on litter decomposition could be different in other ecosystems. Thus, uncovering the influence mechanisms of warming on litter decomposition in desert ecosystems is essential for us to comprehensively understand the impacts of climate change on litter decomposition and carbon cycling.

Soil moisture is another key factor influencing litter decomposition, and winter snow is one of the important sources of soil moisture in desert ecosystems [5]. The changes in winter snow cause considerable changes in soil temperature during the snow-cover period, and highly alter the soil moisture after snow melting [24]. The direct effects of winter snow changes on litter decomposition are the changes in soil hydrothermal conditions, which are closely related to soil microbial activities [25]. For example, previous studies have found that increased snowfall enhanced the activities of soil NAG and α-1,4-glucosidase (AG), which were positively correlated with the litter decomposition rate in temperate forests [25]. And reduced snowfall decreased soil moisture, inhibiting the activities of NAG and AG, which in turn affected the decomposition of cellulose in litter [26] and the catalysis of proteins in litter [27]. Additionally, changes in winter snow cover could cause frequent freeze–thaw cycles, damaging soil microbial growth and activities in the early stage of the snow melting [28]. Notably, most existing studies on the effects of winter snow changes on ecological processes commonly focus on the snow-cover period [13,29,30]. Whether there are legacy effects of winter snow changes (i.e., during the plant growth period after snow melting) on litter decomposition is still unknown and largely overlooked in the existing studies. Furthermore, warming could accelerate the winter snow melting during early spring and intensify the deficiency of soil moisture under the conditions of thicker snowfall during the non-snow-cover period [31]. These two factors not only affect litter decomposition individually but also jointly shape the process of litter decomposition through complex interactions [16,22]. However, the current research on litter decomposition often focuses on the impact of a single climate factor [13,20,29]. Therefore, investigating the potential mechanisms by which warming and winter snow changes affect litter decomposition in desert ecosystems can deepen our understanding of litter decomposition and carbon cycles responding to climate change in arid regions.

Additionally, litter decomposition is also affected by the litter quality, such as the C:N ratio of litter, cellulose content, and degree of fragmentation [32,33]. And previous studies have indicated that litter would decompose faster in their field than in a foreign condition, a phenomenon known as “home-field advantage” [34,35,36]. These differences hinder the comparison of litter decomposition rate across different ecosystem types and disturb the understanding of universally influential patterns of climate change on litter decomposition [37]. Meanwhile, most researchers consider litter falling on the soil surface, and photodegradation is the primary process of litter decomposition in natural ecosystems, especially in arid and semi-arid zones [38,39]. Existing research has indicated that buried litter decomposes faster than the soil surface litter [40], possibly because soil microbes and enzyme activities are involved in the litter decomposition processes [25,33]. Therefore, comparing the differences of litter decomposition in different positions (i.e., soil surface and 2–3 cm belowground) using a standardized substrate can help us precisely quantify the environmental regulation on litter decomposition, excluding the interference of litter-specific properties.

In the present study, to comprehensively dissect the influence mechanisms of warming and winter snow change on litter decomposition, we established an in situ experiment with dynamic-monitoring litter decomposition rate on the soil surface and belowground, soil physicochemical properties, and soil enzyme activities during snow-cover and non-snow-cover periods over five months. Furthermore, we employed standardized litter bags (i.e., red tea, high C:N ratio; and green tea, low C:N ratio), based on the Tea Bag Index method, and litter of dominant plant species in our study site. By doing this, we can make comparisons of decomposition characteristics across different ecosystems at local and global scales [37,41]. Our main objectives were (1) to reveal the potential influence patterns of warming and winter snow changes on litter decomposition; (2) to explore the interactive effects of warming and winter snow changes on litter decomposition; and (3) to compare the litter decomposition patterns across different ecosystem types.

## 2. Results

### 2.1. Effects of Warming and Winter Snow Changes on Soil Physicochemical Properties

Firstly, warming and winter snow changes caused significant changes in soil temperature, soil moisture, and air temperature, but the effects differed between the snow-cover and non-snow-cover periods (Figure 1). During snow-cover period (before 9 March), snow addition increased soil temperature and soil moisture by 5.35% and 3.94%, respectively (Figure 1). Meanwhile, snow removal significantly reduced soil temperature by 3.30%, and highly reduced soil moisture by 5.80% (Figure 1). Warming highly increased soil temperature by 16.17%, particularly under the relatively low air temperature, and it slightly increased soil moisture by 4.58% (Figure 1). Warming increased air temperature by 10.92%, but notably, the warming effects began to be remarkable after 2 February (Figure 1). During non-snow-cover period (9 March to 9 May), snow addition lowered soil temperature by 5.81%, while snow removal increased soil temperature by 0.51% (Figure 1). Obviously, snow addition significantly increased soil moisture by 3.59%, and snow removal highly reduced soil moisture by 6.36% (Figure 1). Warming increased soil temperature by 6.21% and had limited effects on soil moisture, but it caused significant decreases after 23 April. Warming highly increased air temperature by 31.49% (Figure 1c). Across the whole experimental period, the lowest and highest air temperature was −24.5 and 30.2 °C, respectively (Figure 1c). The warming effects stimulated by the OTC caused an average increase of 2.32 °C, in correspondence with the future warming scenario predicted by IPCC [1].

During the snow-cover period, winter snow changes had significant effects on soil NO_3_^−^-N and EC: the snow addition significantly reduced, and the snow removal increased, soil NO_3_^−^-N and EC (Figure 2a; Appendix A). And there were significant interactive effects of winter snow changes and warming on soil EC: warming highly reduced soil EC under the snow-removal treatment (Figure 2a; Appendix A). Meanwhile, warming showed limited effects on soil physicochemical properties (Appendix A). During the non-snow-cover period, all soil physicochemical properties showed an increasing trend compared to the snow-cover period (Figure 2). Specifically, winter snow changes had significant effects on soil NO_3_^−^-N, SOC, and EC (Figure 2b; Appendix A). Soil NO_3_^−^-N and SOC were increased by snow removal, but reduced by snow addition (Appendix A). Meanwhile, soil EC was significantly increased by snow addition, but reduced by snow removal (Figure 2b; Appendix A). Warming significantly increased SOC by 18.1%, especially under snow-addition treatment (Figure 2b; Appendix A). Notably, there were significantly interactive effects of warming and winter snow changes on soil NO_3_^−^-N and SOC (Figure 2b; Appendix A). Warming significantly increased SOC under the snow-addition treatment, but it reduced SOC under the snow-removal treatment (Figure 2b). And warming significantly reduced soil NO_3_^−^-N under the snow-removal treatment, but it exercised limited effects on soil NO_3_^−^-N under the snow-addition treatment (Figure 2b; Appendix A).

### 2.2. Effects of Warming and Winter Snow Changes on Soil Enzyme Activities and Litter Decomposition

Soil carbon (C), nitrogen (N), and phosphorus (P) cycling-related enzyme activities were significantly altered by warming and winter snow changes, and these enzyme activities showed different responses to warming and winter snow changes (Figure 3; Appendix A). During the snow-cover period, warming had significant effects on the activities of soil AG, BG, urease, NAG, and ALP (Appendix A). Warming significantly increased the activities of BG and ALP by 9.3% and 26.2%, respectively (Figure 3a; Appendix A). Winter snow changes also showed significant effects on the activities of soil AG, BG, NAG, and ALP, which clearly increased by 41.2%, 51.43%, 12.95%, and 16.7%, respectively (Figure 3a; Appendix A). Notably, there were significant interactive effects of warming and winter snow changes on these enzyme activities (Appendix A). During the non-snow-cover period, warming significantly reduced the activities of soil NAG and ALP by 22.64% and 20.83%, respectively (Figure 3b; Appendix A). Winter snow changes had significant effects on soil enzyme activities: snow addition increased soil ALP activities by 56.69%, but snow removal reduced the activities of soil AG, BG, and ALP by 39.51%, 14.78%, and 42.66%, respectively (Figure 3b; Appendix A). Also, significant interactions between warming and winter snow changes on these soil enzyme activities were detected (Appendix A). For example, warming increased the activities of soil AG, BG, and NAG under snow-addition treatment, while showing negative effects under snow-removal treatment (Figure 3b; Appendix A).

The decomposition rate of red tea, green tea, and *Erodium oxyrhinchum* exhibited significant differences over time: it showed the lowest decomposition rate in February, and it showed the highest decomposition rate in April (Figure 4; Appendix A). And the decomposition rate of the three litter types which were placed belowground was generally higher than that on the soil surface (Figure 4). Furthermore, warming and winter snow changes had significant effects on the decomposition rates (Appendix A). The snow-addition treatment increased the decomposition rate of red tea, green tea, and *Erodium oxyrhinchum* by 28.07%, 15.8%, and 16.1%, respectively (Figure 4). In contrast, the snow-removal treatment significantly inhibited the decomposition rates with a decrease of 30.82%, 43.9%, and 25.03% for red tea, green tea, and *Erodium oxyrhinchum*, respectively (Figure 4). Warming had stimulatory effects on the litter decomposition rates of the three litter types both on the soil surface and in the belowground location, with an increase of 27.42%, 24.35%, and 45.47% for red tea, green tea, and *Erodium oxyrhinchum*, respectively (Figure 4). Notably, there were significant interactive effects of warming and winter snow changes on the litter decomposition rate of *Erodium oxyrhinchum*: warming highly increased the decomposition rate of *Erodium oxyrhinchum*, especially under the snow-addition treatment (Figure 4). Overall, the litter decomposition rate of green tea was the highest, followed by *Erodium oxyrhinchum*, and the decomposition rate of red tea was the lowest (Figure 4).

In addition, to compare the differences in litter decomposition across different ecosystems, the standard method for monitoring the litter decomposition rate, i.e., TBI parameters, was used, and the related published data were collected in our study (Figure 5). The results revealed that the decomposition rate constant (k) was the highest at the lowland tropical forest in Panama (Site 2), gradually decreased with the declines of precipitation, and reached the lowest level at our study site (Gurbantünggüt Desert in Xinjiang; Site 12) (Figure 5; Appendix A). In contrast, the stability factor (S) was the lowest at the geothermal wet grassland warmed in Iceland (Site 1), but it was the highest at our study site (Gurbantünggüt Desert in Xinjiang; Site 12). These results indicate that precipitation and temperature are the top two main factors controlling litter decomposition.

### 2.3. Underlying Mechanisms of Warming and Winter Snow Changes on Litter Decomposition

Having demonstrated how the combination of warming and winter snow changes had significant effects on the litter decomposition, we further wanted to identify the potential influence pathways of warming and winter snow changes on the litter decomposition of the three litter types at the two survey points. Therefore, we focused on analyzing the drivers influencing these two factors based on random forest analysis and structural equation models. The random forest analysis indicated that soil enzyme activities were the main drivers of the alterations seen in the litter decomposition. During the snow-cover period, soil BG and AG activities were the main drivers of the three litters’ decomposition. Specifically, soil BG activity contributed 27.2%, 21.83%, and 47.5% to the decomposition of the red tea, green tea, and *Erodium oxyrhinchum*, respectively (Figure 6a–c). Soil AG activity contributed 19.3% and 34.2% to the decomposition of the red tea and *Erodium oxyrhinchum*, respectively (Figure 6a,c). Furthermore, we also found that soil EC and NO_3_^−^-N also significantly contributed 17.8% and 17.37% to the decomposition of the red tea, respectively (Figure 6a). In contrast, during the non-snow-cover period, soil enzyme activities were also the main drivers of the alterations seen in the litter decomposition, but they were the ALP and NAG, and not the BG and AG. Specifically, soil alkaline phosphatase activity contributed 28.9%, 29.1%, and 44.6% to the decomposition of the red tea, green tea, and *Erodium oxyrhinchum*, respectively (Figure 6d–f). Meanwhile, soil NO_3_^−^-N was also the main influence driving litter decomposition, with a contribution rate of 34.2% and 22.6% to the decomposition of red and green tea, respectively (Figure 6d,e). Furthermore, soil BG and NAG activities, and EC also, significantly affected red tea litter decomposition, and their contribution rates were 27.3%, 26.1%, and 21.9%, respectively (Figure 6d).

Finally, structural equation models were constructed to reveal the potential influencing pathways of warming and winter snow changes on the decomposition of the three litter types during the snow-cover and non-snow-cover periods, respectively. During snow-cover period, the best SEM model explained 72%, 69%, and 44% of the variations in the decomposition rate of red tea, green tea, and *Erodium oxyrhinchum*, respectively (Figure 7a). Consistent with the results of random forest analysis, soil BG activities were significantly and negatively related to the decomposition rate of green tea and *Erodium oxyrhinchum*, and soil AG activity was significantly and positively related to the decomposition rate of red tea (Figure 7a). Winter snow changes indirectly increased decomposition rate of green tea and *Erodium oxyrhinchum* by reducing soil BG activity, which was negatively related to the decomposition rate of these two litter types (Figure 7a). Notably, both warming and winter snow changes had direct priming effects on the decomposition rate of these three litter types (Figure 7a). In contrast, during the non-snow-cover period, winter snow changes influenced soil BG and NAG activities by positively increasing soil EC and negatively decreasing soil NO_3_^−^-N content (Figure 7b). Winter snow changes increased the decomposition rate of red tea, green tea, and *Erodium oxyrhinchum* by increasing soil ALP activities (Figure 7b). Meanwhile, winter snow changes increased the decomposition rate of green tea by decreasing soil BG activities, which were negatively related to the decomposition rate of green tea (Figure 7b). Warming promoted the decomposition rate of the three litter types mainly through the direct pathways (Figure 7b). Furthermore, this SEM results clearly revealed that winter snow changes had significant legacy effects on litter decomposition after the snow had melted, and the effects were even higher than the effects of warming on litter decomposition (Figure 7b).

## 3. Discussion

### 3.1. Potential Mechanistic Drivers of Litter Decomposition Rates Under Winter Snow Changes

Precipitation and temperature are the major factors influencing litter decomposition across different ecosystems [11,22,34]. Scarce precipitation and high temperature are the typical climatic characteristics of desert ecosystems [23]. Winter snow changes caused notable changes in soil temperature and moisture, and subsequently impacted litter decomposition. Our results showed that increased snow cover promoted the decomposition rate of three litter types, as is consistent with previous studies [42,43]. This may be due to two reasons: Firstly, the changes in the soil hydrothermal conditions caused by the winter snow changes could be the main reason [42]. In the present study, snow addition raised soil temperature by 6.5% during the snow-cover period, and it increased soil moisture by 3.7% due to the increased snow cover of the previous years. Notably, snow addition highly reduced soil temperature due to the high increases in soil moisture after the snow melted during the non-snow-cover period with high air temperature. The reduced soil temperature with high soil moisture could be conducive to litter decomposition, and it could be the reason that increased snow cover had the most notable effects on litter decomposition in April.

Secondly, favorable soil hydrothermal conditions could promote soil microbial activities with high soil enzyme activities, especially in the early spring, when the soil temperature was low, and in late May, when soil temperature was high [42,44]. Our study found that snow addition enhanced soil enzyme activities, which were the main factors driving the alterations in litter decomposition. These results were consistent with previous studies [9,13,42]. In one subtropical natural forest, warming reduced litter decomposition rate by inhibiting soil enzyme activities, such as soil BG and acid phosphatase [9]. And one previous study indicated that climate warming not only influenced the soil carbon-degrading enzymes but also altered the soil C:P enzyme ratios, subsequently impacting litter decomposition [13]. These results highlight the critical roles of soil enzyme activities in mediating the litter decomposition processes. It is worth noting that soil enzymes related to the carbon cycle (i.e., AG and BG) exercised significant effects on litter decomposition during the snow-cover period, while soil enzymes related to phosphorus and nitrogen cycles (ALP and NAG) controlled the litter decomposition during the non-snow-cover period. This result is in line with the findings in the Tibetan Plateau showing that, during the early growth season, warming reduced soil carbon-degrading enzymes (e.g., BG), but it highly increased soil phosphorus-related enzymes (e.g., ALP) [13]. This is possibly due to the fact that soil enzyme activities reflect the relative nutrient limitations of soil microbial nutrient assimilation and growth [14,45]. One recent study revealed that, during snow-cover period, soil carbon was the primary limiting factor for microbial growth, while N and P became the major limiting factors during the plant growth season, resulting in the different roles of soil carbon-, nitrogen-, and phosphorus-cycling enzymes [46].

Furthermore, our study indicated that snow removal inhibited litter decomposition; the effects were still observed even during the non-snow-cover period, as was consistent with the previous studies [25,47]. This may be because that soil temperature was significantly reduced with no snow cover in winter, and soil moisture was much lower than that in the ambient winter snow treatment after the snow had melted. Notably, contrary to our expectation, we observed a positive effect of decreased snow cover on soil enzyme activities during the snow-cover period. A possible reason could be that soil temperature and moisture were relatively low, leading to an increase in the frequency of freeze–thaw cycles, and subsequently accelerated the degradation of soil organic matter [48]. This process released large amounts of organic materials and increased soil nutrient contents, as indicated in our study. This provided nutrients for soil microorganisms, and promoted their growth and metabolic activities, which in turn increased soil enzyme activities [14,42]. Meanwhile, the increases in soil BG activity were indicated to decrease the litter decomposition rate of green tea during the snow-cover period, which was contrary to the positive effects of soil enzyme activities on litter decomposition [13]. This may be because the enzyme chemometrics, requiring synergistic action of multiple enzymes on litter decomposition [11,47]. For example, soil BG is a crucial enzyme in cellulose saccharification [49]. When soil BG activity was high, more glucose was produced, possibly leading to an accumulation of cellobiohydrolases (CBHs) during the cellulose decomposition process. The accumulation of glucose and CBHs would in turn inhibit the soil BG activities, especially in low-temperature environment [49]. Notably, although changes in soil enzyme activities can represent the variations in soil microbial community and functions, we did not directly test the changes in soil microbial community which were related to soil enzyme activities. Together, these results highlighted that winter snow changes exercised significant effects on litter decomposition, and there were notable legacy effects of winter snow changes on litter decomposition. Therefore, future studies considering the legacy effects of winter snow change on litter decomposition in the field and the changes in soil microbial community are needed to draw more rigorous conclusions about the litter decomposition process in response to winter snow changes.

### 3.2. Potential Mechanisms Underlying Warming-Induced Changes in Litter Decomposition

Warming is one of the most serious issues facing our earth, and it is considered to be the primary factor controlling litter decomposition through its direct influences on soil microbial activities across different ecosystems [12,20]. Our study revealed that warming promoted litter decomposition, especially during the non-snow-cover period, as is consistent with a large variety of ecosystems in previous studies [12,15,20]. It is widely accepted that the positive effects of warming on litter decomposition are directly by accelerating soil enzyme activities [12,50]. Our results supported this explanation under the ambient snow treatment. Notably, our results further indicated that the promoting effects of warming on the litter decomposition were also related to the soil NO_3_^−^-N content. Our results showed that warming significantly reduced soil NO_3_^−^-N content. On one hand, this may be because warming could show an inhibitory effect on the activity of soil nitrifying bacteria, thereby reducing the conversion of NH_4_^+^-N to NO_3_^−^-N [51]. On the other hand, it may be because a large amount of available nitrogen was absorbed by enhanced plant growth during the growth season [52]. The declines in soil NO_3_^−^-N content will in turn stimulate soil microbes to synthesize enzymes to acquire more nitrogen, meeting their metabolic needs [53]. Meanwhile, there are existing studies indicating that warming shows minimal and negative effects on litter decomposition, possibly due to the high natural environmental temperature and limited soil moisture in the studied area [11].

Interestingly, our results revealed significant interactions of warming and winter snow changes on soil enzyme activities and litter decomposition, indicating that soil moisture was likely the other main driving factor influencing litter decomposition in the studied desert ecosystem. These results were inconsistent with the findings, which were based on one subtropical ecosystem [15,54]. In the subtropical ecosystem, warming exercised limited effects on litter decomposition, and even its negative effects were limited [55]. Furthermore, in the subtropical and tropical ecosystems, soil moisture is proven to unlikely be the main factor influencing litter decomposition [15]. Meanwhile, our results indicated that changes in soil moisture induced by winter snow changes were the other factors influencing litter decomposition in the harsh environment. Water is the primary factor that significantly impacts ecological processes, especially in arid and semi-arid areas [23]. Previous studies have revealed that increases in soil moisture promote soil enzyme activities and subsequently increase litter decomposition rate [38,43]. Our results indicated that increased snow cover significantly raised soil moisture at the early stage of the snow melting, and warming advanced the snowmelt, creating favorable conditions with moderate soil temperature and moisture for soil microbes. This could be the reason why the highest litter decomposition rate was in April, thus highlighting the important roles of soil moisture in altering litter decomposition, which was consistent with the previous studies conducted in water-limited ecosystems [6,21]. Furthermore, our results found that the synergistic effects of warming and winter snow changes on litter decomposition were through increasing soil alkaline phosphatase activity. This may be because soil phosphorus was the major nutrient-limiting factor for diverse ecological functions in the study area; when soil hydrothermal conditions became better, soil microbes could preferentially produce more alkaline phosphatase to reduce the phosphorus limitation compared to the other nutrients [46]. Overall, our results revealed that warming and winter snow changes exercised synergistic effects on litter decomposition and promoted litter decomposition in the desert ecosystem, and the influence pathways were different with other ecosystem types. Special attention should be paid to understanding the effects of multiple climate changes on litter decomposition to better evaluate the effects of climate change on soil carbon cycling.

### 3.3. Comparative Analysis of TBI Parameters Across Different Ecosystems

Perhaps the most important aspect of our study that influenced out findings is that we compared the litter decomposition rate of different ecosystems based on the standard method, the Tea Bag Index method. Traditional litter decomposition studies make it difficult to compare the litter decomposition rate across different ecosystems due to the differences in plant chemical composition (e.g., lignin and cellulose content) [37,56]. Previous studies have indicated that the initial quality between forest litter and grassland litter could mask the true regulatory effects of climate change on litter decomposition due to the “home-field advantage” [34,37]. The Tea Bag Index method addresses this issue by using standardized materials (i.e., green tea and red tea bags) with strictly homogenized chemical compositions and physical structures [37,41]. Based on this method, we found that there were clear differences in k and S between different ecosystems, and these measures gradually changed along the gradient of temperature and precipitation. Compared to the litter decomposition rate (k) of forests, swamps, grasslands, and pastures in other countries, the k value of the Gurbantünggüt Desert (our study site) in China is the lowest. This may be due to several reasons. First, the Gurbantünggüt Desert is located in an inland area with an extremely arid climate and scarce annual rainfall, resulting in very low soil moisture and inactive soil microbial activities [23,57]. Second, the region experiences extreme day–night temperature differences, with high temperatures during the day and low temperatures at night. These extreme temperature conditions are also unfavorable for soil microbial growth and reproduction, further affecting the efficiency of organic-matter decomposition [23]. In contrast to the k value, our study observed that the stabilization factor (S) in the Gurbantünggüt desert was the highest. The high S value indicated that soil organic carbon is more resistant to further decomposition, making it easier to be stabilized and form stable soil organic carbon pools. Among the twelve study sites, the S value increased with an increase in the soil carbon sequestration potential of terrestrial ecosystems. The S value was lower in forests (Site 2), moderate in humus-rich forests (Site 3), and higher in carbon-accumulating peatlands (Sites 7 and 8). Furthermore, a comparison of the S values between warm grasslands and ambient grasslands in Iceland (Sites 1 and 5) showed that the S value is lower in warm grasslands, indicating a positive correlation between reduced carbon storage and increased temperature. These results indicate that soil organic carbon may be less stable in environments with higher temperatures and greater precipitation, while in arid regions, it is more conducive to stable soil carbon storage.

Additionally, our results found a negative effect of “home-field advantage” on litter decomposition rate, indicated by the higher decomposition rate of green tea than that of *Erodium oxyrhinchum*. Previous studies have indicated that “home-field advantage” can occur due to highly contrasting litter qualities and single dominating plant species [58,59]. Furthermore, the effects of “home-field advantage” on litter decomposition also highly depend on soil properties, especially soil water availability [36]. Due to the high diversity of the plant community in the study area, less contrasting litter quality between green tea and *Erodium oxyrhinchum*, and low soil water content, “home-field advantage” was not evident, and neither were negative effects. Furthermore, litter could be consumed by soil animals, increasing the fragmentation degree, and subsequently accelerating the decomposition rate [60]. Our results found that the decomposition rate of all three litter types underground were higher than that at the soil surface, which was contrary to the findings revealing that UV radiation exercised major effects on litter decomposition [7]. Our results highlight the key roles of soil microorganisms in litter decomposition. Notably, litter decomposition processes can be impacted by soil animals, such as nematodes, mites, and springtails, which can accelerate litter decomposition by increasing litter surface area [61]. One previous study indicated that the mesofauna and macro-soil animals exercised limited effects on litter decomposition [62]. This suggests that the effects of soil animals on litter decomposition may differ across different litter types and ecosystems. Thus, although the same mesh size was used to prevent the roles of soil animals for the three litter types in our study, further studies should be performed considering the effects of soil animals on litter decomposition. Also, our results only focused on a limited number of time points of litter decomposition, and they did not reveal the relationships between litter decomposition and soil carbon pools. Therefore, long-term continuous observations focusing on the relationships between litter decomposition and soil carbon pools are imperative for a comprehensive understanding of litter-decomposition and carbon-cycling responses to global climate change.

## 4. Materials and Methods

### 4.1. Study Site

This study was carried out at the southern edge of the Gurbantünggüt Desert (34°09′–49°08′ N, 73°25′–96°24′ E), which is the second-largest desert, located in Northwestern China, Central Asia. The Gurbantünggüt Desert (4.88 × 10^4^ km^2^) is characterized by a continental arid temperate climate with long winters and short springs. This region has an annual average temperature that ranges between 4 and 7 °C, an annual average precipitation of 215.6 mm, and an average growing-season precipitation level (March to May) of 67.9 mm [63]. However, the annual average potential evapotranspiration exceeds 1590 mm [57]. During winter, this region is typically covered by snow, with a depth of approximately 30 cm. Snowmelt and spring precipitation, which account for approximately 65% of the annual precipitation, serve as the primary water sources to provide sufficient soil water for the germination of ephemeral plant seeds and finish their life cycle [57]. The herbaceous vegetation is primarily composed of ephemeral plants, with *Erodium oxyrhinchum* being the dominant species [64]. The soils are gray desert soils (Chinese classification) with a high soil pH of 9.55 ± 0.14, low fertility, and high bulk density of 1.58 ± 0.1 g/m^3^ [65].

### 4.2. Experimental Design

The experiment was established in early January 2022 using a split-plot randomized block design with two factors, i.e., warming and winter snow changes. We set “winter snow changes” as the main plot (each 2 m × 4 m in size), and “warming” as the sub-plot (each 2 m × 2 m in size; Appendix A), with 1 m buffer zones between adjacent main plots and 0.2 m buffer zones between adjacent sub-plots. For winter snow changes, we included three levels, i.e., ambient winter snow (control, CK), 100% snow removal (−S), and 100% snow addition (+S). A snow-shielding board was installed 20 cm above the ground for the snow-removal treatment to achieve 100% snow removal. The snow above the snow-shielding board was uniformly redistributed to the snow-addition treatment after each snowfall event (almost once a month). The snow was redistributed until the cessation of snowfall (normally at the end of February each year) to achieve 100% snow addition. The snow in the ambient winter snow (CK) was undisturbed. For the warming factor, we included two levels, i.e., ambient temperature (control, CK) and warming (+T). Open top chamber (OTC) was used to simulate warming continuously. The OTC was square, measuring 2 m long, 2 m wide, and 80 cm high, and it was expected to increase air temperature by 1.5–2 °C in our study. The OTC size can cover the patchy desert vegetation and the spatial heterogeneity of soil microbes. Overall, six treatments with five replicates (five blocks) for a total of 30 sub-plots were included in the present study. Soil temperature and moisture probes were placed in each experimental plot within one block to monitor the changes in soil temperature and moisture at the 0–10 cm and 10–20 cm soil depths. An air temperature-monitoring probe was placed 10 cm above the ground inside and outside the OTC to monitor the warming effects of OTC on air temperature. Notably, based on the air temperature, soil temperature, soil moisture, and snowmelt situations, we divided our experimental period into two sub-periods, which revolved around 9 March, i.e., before 9 March, named “snow-cover period”, and after 9 March, named “non-snow-cover period” (Figure 1).

### 4.3. Litterbag Placement and Collection

Following the standardized protocol by the Tea Composition initiative on assessing litter decomposition, we used two tea types with different C:N ratios, i.e., Lipton green tea and Lipton rooibos tea (red tea). The tea bag contained 2 g tea, with the green tea bag consisting of 89% green tea and the red tea bag consisting of 93% red tea. The tea bags had a mesh size of 0.25 mm, which allowed soil microorganisms to enter while excluding soil large fauna [37]. Furthermore, the litter of the dominant herbaceous plant species (*Erodium oxyrhinchum*) was also selected to reveal the effects of “home-field advantage” on the litter decomposition and compare the different decomposition rates of different litter quality in our study. We collected *Erodium oxyrhinchum* in the same experimental area, and 2 g of *Erodium oxyrhinchum* was weighed and placed into 6.5 × 4.5 cm nylon bags, which were similar to the tea bags. Notably, the C:N ratio of red tea, green tea, and *Erodium oxyrhinchum* is approximately 43, 12, and 3, respectively [37]. The C:N ratios represented an ideal gradient (i.e., high, medium, and low) to uncover the effects of warming and winter snow changes on different litter quality. Additionally, to compare the effects of different litter locations on the decomposition rate, on the soil surface, we simulated the actual litterfall conditions, and 3 cm belowground was chosen as our focal point for the soil microbial effects. The litter bags of green and red tea were vertically inserted into the soil, and litter bags of dominant species were inserted at an angle into the soil to ensure within 3 cm soil layer. In total, 180 *Erodium oxyrhinchum* litter bags, green tea bags, and red tea bags (6 treatments × 2 positions × 3 parallels × 5 replicates) were placed on 9 January 2023. All litter bags were harvested once a month, and the same number of three litter bags were replaced. On 9 May 2023, we finished the final harvest, and there were a total of 2160 litter bags with the three litter types and four collective points. Before burying the litter bags, they were oven-dried at 70 °C for 48 h until reaching a constant weight, and the initial weight was recorded. After retrieval, each litter bag was cleaned to remove soil and roots, and dried again at 70 °C until constant weight; mass loss at each time interval was determined. The decomposition rate constant (k), which is an indicator of the initial decomposition speed of litter, was calculated for the litter of red tea, green tea, and *Erodium oxyrhinchum*. It was estimated using the modified Olson exponential decay model, [66,67], $M_{t}/M_{0}=e^{-kt}$. In the formula, M_t_ represents the residual mass (g) of litter after decomposition for a period t, M_0_ is the initial mass (g) of the litter, t is the decomposition time expressed in months, and k is the decomposition rate constant. A larger k value indicates a faster decomposition rate, while a smaller k value indicates a slower decomposition rate. The stabilization factor (S) measures the proportion of unstable components in the litter that eventually stabilize and become recalcitrant, transforming into soil organic matter. It was calculated by the following equation: S = 1 − *M*/$M_{t}$, where a is the decomposable of green tea [37]. A larger S value indicates a lower decomposition rate and inhibition of decomposition. More detailed information can be found at http://www.teatime4science.org (accessed on 11 January 2022), which is provided by the Tea Bag Index (TBI) research group.

### 4.4. Soil Sampling and Determination of Soil Physicochemical Properties and Enzyme Activities

Soil samples were collected twice: once after the snow had melted, on 9 March (the second litter collection point), and again at the end of the growing season, on 9 May (the final litter harvest point). Soil cores (5 cm diameter; 10 cm height) were collected from three random locations in each experiment plot, and then sieved to 2 mm to mix thoroughly into one sample in situ. Subsequently, each soil sample was divided into three sub-samples: One sub-sample was placed in a shaded area to air dry for determination of soil organic carbon (SOC), pH, and electrical conductivity (EC). The second sub-sample was stored in −20 °C for the measurement of ammonium nitrogen (NH_4_^+^-N) and nitrate nitrogen (NO_3_^−^-N). The third sub-sample was stored in 4 °C for analyzing soil enzyme activities related to soil carbon, nitrogen, and phosphorus cycling, i.e., α-1,4-glucosidase (AG) and β-1,4-glucosidase (BG), representing organic matter degradation; urease (Urea) and β-1,4-N-acetylglucosaminidase (NAG), representing the hydrolysis of nitrogen-containing organic compounds; and alkaline phosphatase (ALP), representing the conversion of organic phosphorus to inorganic phosphorus.

Soil organic carbon was determined using the oxidation method with a potassium dichromate–sulfuric acid solution. Soil pH and EC were measured in a 1:5 (*w*:*v*) water solution with a pH meter (Seven Easy, Mettler-Toledo, Greifensee, Switzerland) and a conductivity meter (S20 K, Seven Easy, Mettler-Toledo, Greifensee, Switzerland), respectively. Soil NH_4_^+^-N and NO_3_^−^-N were extracted with 2 M KCL, and their contents were determined with a continuous flow analyzer. Soil enzyme activities were determined by the 96-well microplate fluorescence method with the standard MUB (4-methylumbelliferone)-linked substrates. Specifically, 1 g of fresh soil was homogenized in 100 mL of 50 mmol·L^−1^ sodium acetate buffer via vortex mixing for 5 min. Then, 200 μL of the soil slurry and 50 μL of substrate solution were pipetted into 96-well microplates and incubated in the dark at 25 °C for 4 h. Potential enzyme activity was quantified using a SpectraMax M5 microplate reader (Molecular Devices, Sunnyvale, CA, USA), and fluorescence was measured with excitation at 365 nm and emission wavelengths at 450 nm.

### 4.5. Statistical Analysis

At first, all measured parameters were pre-tested for homogeneity of variance and normal distribution; no variance inhomogeneity was detected, and the data were normally distributed. Two-way ANOVA was following employed to test the main and interactive effects of warming and winter snow changes on soil physicochemical properties (i.e., NH_4_^+^-N, NO_3_^−^-N, pH, EC, and SOC; Appendix A) and soil enzyme activities (i.e., AG, BG, urease, NAG, and ALP; Appendix A) after the snow had melted and at the end of the growing season, respectively. Meanwhile, three-way ANOVA was performed to test the main and interactive effects of warming, winter snow changes, and monitoring time on the litter decomposition rate (k) of red tea, green tea, and *Erodium oxyrhinchum* for the soil surface and underground location, respectively (Appendix A). Secondly, heat map was used to present the soil physicochemical properties (Figure 2). Because the order of values of the individual properties differed substantially, we standardized all properties to a common scale, ranging from 0 to 1, i.e., standardized value = (X − X_min_)/(X_max_ − X_min_), where X, X_max_, and X_min_ are the target variable, its maximum value, and minimum value across all samples, respectively, which are widely used in the previous studies [68,69]. Additionally, a random forest model was used to evaluate the relative contributions of soil physicochemical properties and soil enzyme activities to the decomposition rate of three different litter types (i.e., red tea, green tea, and *Erodium oxyrhinchum*) after the snow had melted and at the end of the growing season, respectively. Finally, structural equation model (SEM) was applied to reveal the potential influence pathways of warming and winter snow changes on litter decomposition rate after the winter snow had melted and at the end of the growing season, respectively. The fitness of the model was evaluated using the χ^2^ test, root-mean-square error of approximation (RMSEA), goodness-of-fit index (GFI), and Akaike information criterion (AIC) with *p*-value > 0.05 [70]. Statistical analyses were performed using SPSS 22 (SPSS Inc., Chicago, IL, USA), and graphs were created using Origin 2022 software (version 9.9, OriginLab Corporation, Northampton, MA, USA). Random forest analysis and SEM analysis were conducted in R version 3.4.1, using the “rfPermute” and “lavaan” packages, respectively.

## 5. Conclusions

We conclude that warming and increased snow cover increased the litter decomposition rate of red tea, green tea, *Erodium oxyrhinchum*, and had significant synergistic effects on litter decomposition. The effects of warming and winter snow changes on litter decomposition differed between snow-cover and non-snow-cover periods, and the effects were more pronounced in April, when the hydrothermal conditions were the best. The decomposition rates of all three litter types underground were higher than their decomposition rates on the soil surface, highlighting the important roles of soil microbes in accelerating litter decomposition. Furthermore, we found that warming and winter snow changes influenced litter decomposition through different pathways during snow-cover and non-snow-cover periods; that is, warming and winter snow changes altered litter decomposition by influencing soil enzyme activities related to soil carbon cycling during the snow-cover period, while influencing soil enzyme activities related to soil phosphorus cycling during the non-snow-cover period. And, notably, decreased snow cover promoted soil enzyme activities during snow-cover period, and the effects of winter snow changes on litter decomposition were more pronounced after the snow had melted (i.e., legacy effects). More interestingly, a cross-ecosystem comparison based on the Tea Bag Index method showed that the decomposition rate (k) was the lowest, and the stability factor (S) was the highest in the Gurbantünggüt Desert. These results highlight the critical roles of temperature and precipitation on litter decomposition, and indicate that although soil carbon input in desert ecosystems is slow due to the low litter decomposition, the ability to form stable organic carbon is high. Whether there is a relationship between litter decomposition rate and soil carbon pools remains an open question. Future research considering long-term monitoring, soil macro-animals, and soil carbon sequestration should be performed to predict the long-term effects of climate change on litter decomposition and inform evidence-based policies aimed at improving desert soil carbon sink capacity in the face of global climate change.

## Figures and Tables

**Figure 1 plants-14-02741-f001:**
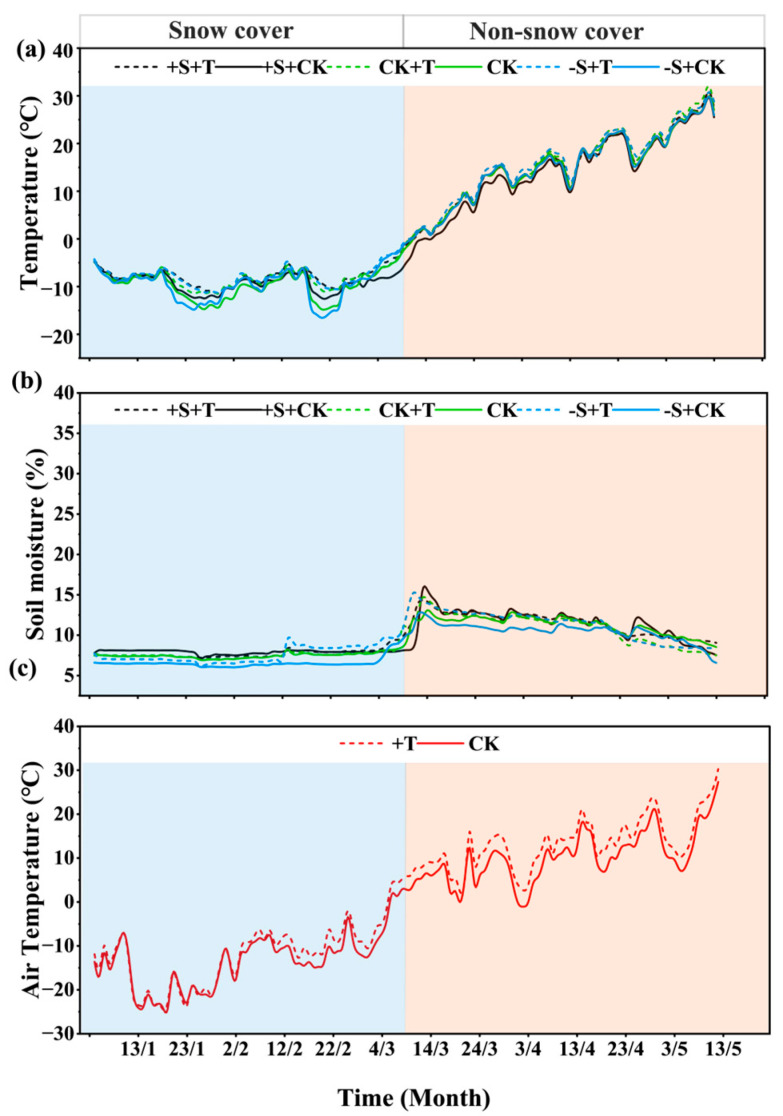
Changes in soil temperature (**a**), soil moisture (**b**), and air temperature (**c**) under warming and winter snow changes during experimental period (January to May 2024). The snow-cover and non-snow-cover period was divided by 9 March, when winter snow began to melt. Black line represents 100% snow-addition treatment (+S), green line represents ambient winter snow treatment (CK), blue represents 100% snow-removal treatment (−S), solid line represents ambient temperature treatment (CK), and dashed line represents warming treatment (+T).

**Figure 2 plants-14-02741-f002:**
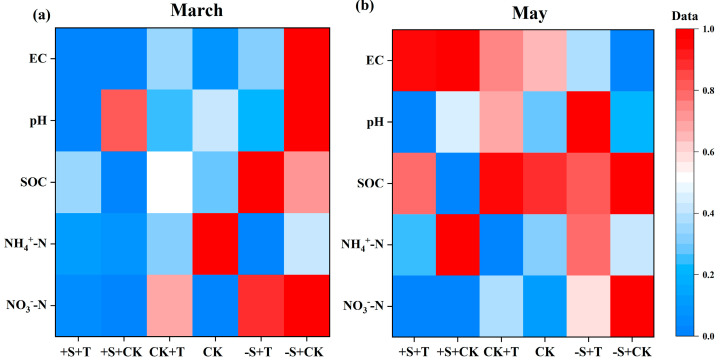
Heat map illustrating the effects of warming and winter snow changes on soil physicochemical properties (i.e., EC, pH, SOC, NH_4_^+^-N, and NO_3_^−^-N) after the snow melted ((**a**), March) and at the end of the growing season ((**b**), May), respectively. +T, warming; CK, ambient temperature; +S, 100% snow addition; CK, ambient winter snow; −S, 100% snow removal. EC, electrical conductivity; SOC, soil organic carbon; NH_4_^+^-N, ammonium nitrogen; NO_3_^−^-N, nitrate nitrogen. The values of these properties were standardized to a common scale ranging from 0 to 1 (see Section 4). For original data, see Appendix A.

**Figure 3 plants-14-02741-f003:**
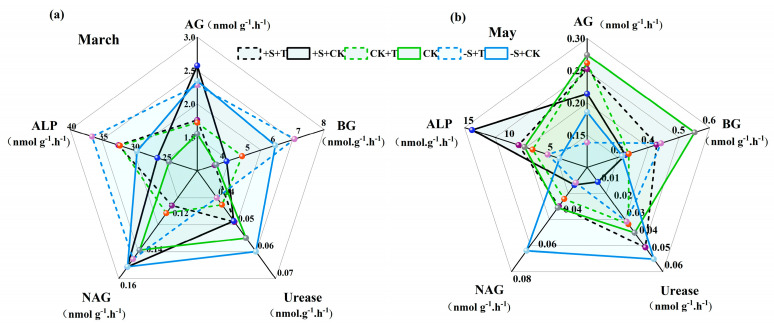
Impact of warming and winter snow changes on soil enzyme activities after the snow melted ((**a**) March) and at the end of growing season ((**b**) May), respectively. AG, α-glucosidase; BG, β-glucosidase activity; NAG, β-1,4acetamido-glucosidase; ALP, alkaline phosphatase. Black line represents 100% snow-addition treatment (+S), green line represents ambient winter snow treatment (CK), blue represents 100% snow-removal treatment (−S), solid line represents ambient temperature treatment (CK), and dashed line represents warming treatment (+T).

**Figure 4 plants-14-02741-f004:**
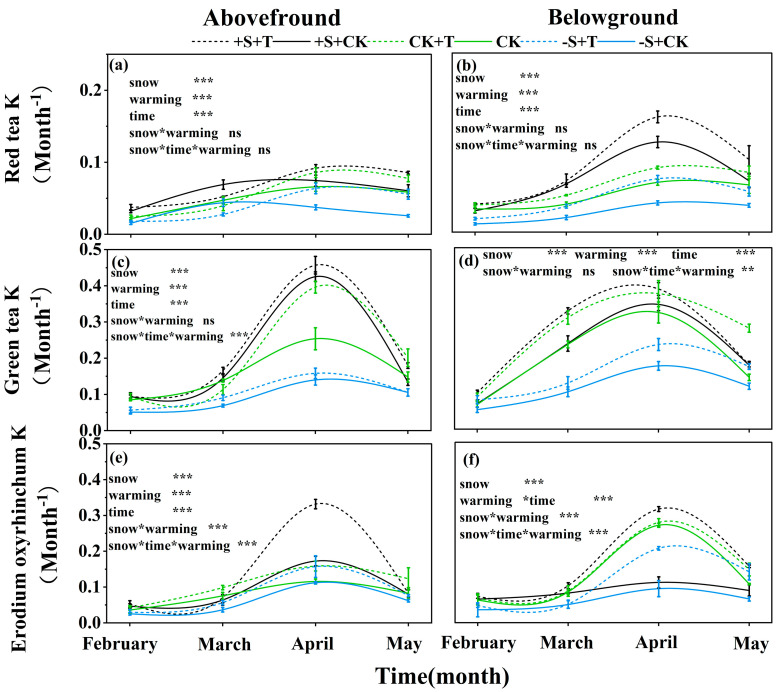
Impact of warming and winter snow changes on litter decomposition rate (k) of red tea (**a**,**b**), green tea (**c**,**d**), and *Erodium oxyrhinchum* (**e**,**f**) at the soil surface and belowground, respectively. Black line represents 100% snow-addition treatment (+S), green line represents ambient winter snow treatment (CK), blue represents 100% snow-removal treatment (−S), solid line represents ambient temperature treatment (CK), and dashed line represents warming treatment (+T). ***, *p* < 0.001; **, *p* < 0.01; *, *p* < 0.05; ns, *p* > 0.05.

**Figure 5 plants-14-02741-f005:**
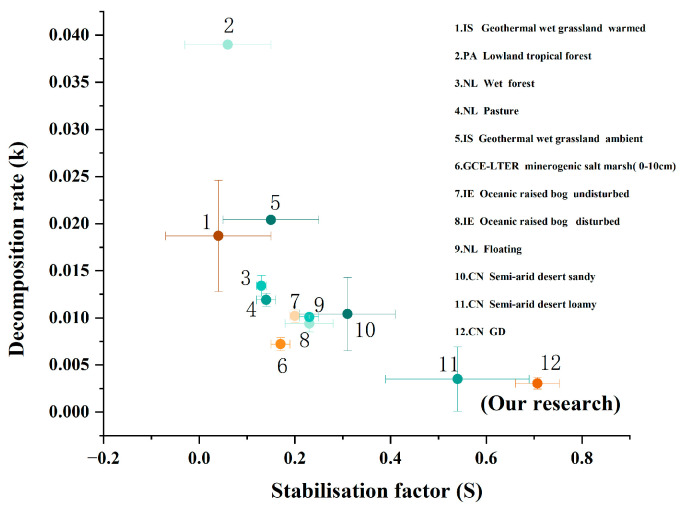
Decomposition rate (k) and stabilization factor (s) at different sites, showing the differences between and within ecosystems based on the Tea Bag Index method. CN, China; NL, The Netherlands; IS, Iceland; GD, Gurbantünggüt Desert. Detailed information is given in Appendix A.

**Figure 6 plants-14-02741-f006:**
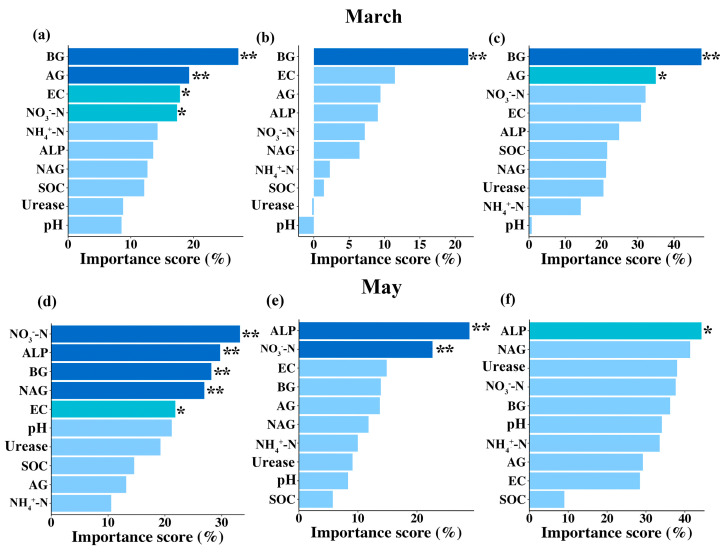
Contribution of soil physicochemical properties and soil enzyme activities to the litter decomposition rate of red tea, green tea, and *Erodium oxyrhinchum* after the snow had melted ((**a**–**c**) March) and at the end of growing season ((**d**–**f**) May), based on random forest analysis, respectively. Importance of the contribution is divided into three classes: dark blue, ** *p* < 0.01; sky blue, * *p* < 0.05; and light blue, non-significant, *p* > 0.05.

**Figure 7 plants-14-02741-f007:**
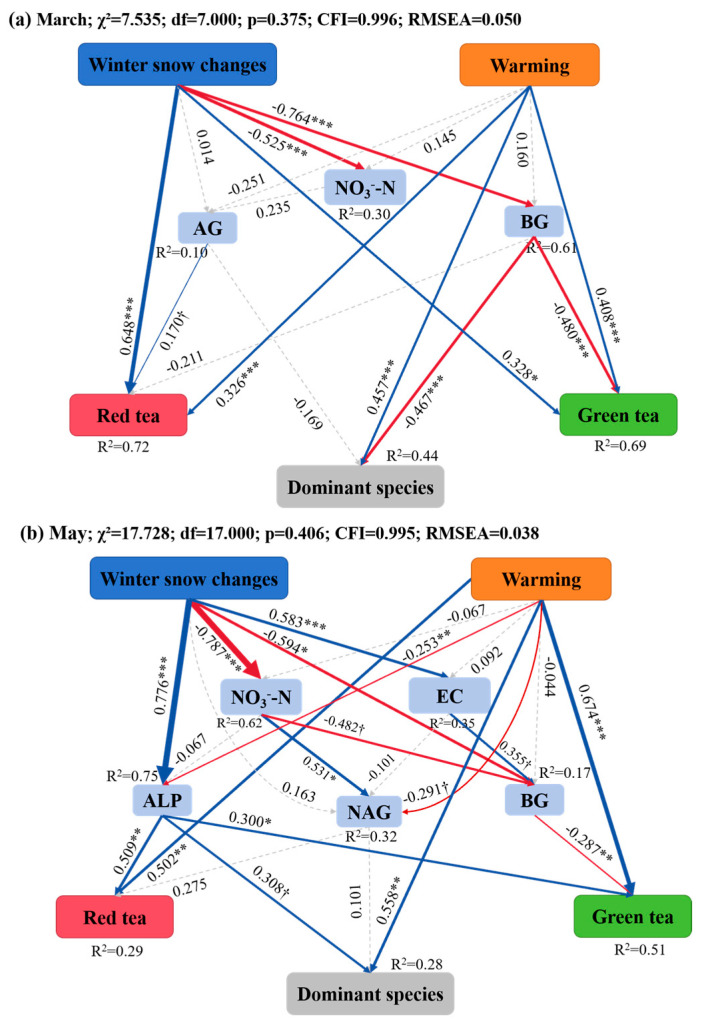
Structural equation modeling (SEM) illustrates the causal pathways through which warming and winter snow changes affected the litter decomposition rate after the snow melted ((**a**) March) and at the end of the growing season ((**b**) May), respectively. Solid red and blue arrows indicate negative and positive significant effects, respectively, and gray dashed lines indicate non-significant effects. Numbers above the arrows indicate the magnitude of the standardized SEM coefficients (^†^ *p* < 0.1, * *p* < 0.05, ** *p* < 0.01, and *** *p* < 0.001). R^2^ values indicate the proportion of variance explained by each variable.

## Data Availability

Data are contained within this article and the Appendix A.

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
