# Peer review of "Bidirectional Regulatory Effects of Warming and Winter Snow Changes on Litter Decomposition in Desert Ecosystems"

_plants, 2025, doi:10.3390/plants14172741_

Round 1
Reviewer 1 Report
Comments and Suggestions for Authors
My review
This study investigates the interactive effects of warming and altered winter snow cover on litter decomposition in the Gurbantunggut Desert, employing a well-designed field experiment and standardized "Tea Bag Index" methodology. The manuscript presents novel insights into desert ecosystem responses to climate change, particularly regarding soil enzyme activity and carbon cycling. The work is timely, methodologically rigorous, and addresses a critical knowledge gap in arid land biogeochemistry. However, some aspects require clarification to strengthen the conclusions.
Specific Comments
1 Material Transparency: The OTC material (typically transparent polycarbonate or glass) is not specified, which may affect light transmittance and humidity. Revision suggestion: Add OTC photos or schematic diagrams in the supplementary materials (e.g., Supplementary Figure S1).
2 Spatial Representativeness: Is a 2 m × 2 m OTC sufficient to represent patchy desert vegetation? The authors need to justify whether this size adequately covers the spatial heterogeneity of soil microbes.
3 Warming Method: Is the bottom of the warming device open? To my knowledge, OTCs usually cause soil cooling (negative warming), but this paper reports positive warming—why is that?
4 Abbreviation Usage: "TBI" only appears in the Methods section but should be spelled out in full upon its first mention in the text.
5 Color Scheme in Figures: The shades of sky blue and light blue in the figures are difficult to distinguish.
6 Figure 2 Heatmap: The heatmap could convey more information. Typically, heatmap values can range from -1 to +1 to reflect positive/negative effects, with color intensity indicating the magnitude of change. Asterisks could also be added to denote significance.
Author Response
Comments 1: This study investigates the interactive effects of warming and altered winter snow cover on litter decomposition in the Gurbantunggut Desert, employing a well-designed field experiment and standardized "Tea Bag Index" methodology. The manuscript presents novel insights into desert ecosystem responses to climate change, particularly regarding soil enzyme activity and carbon cycling. The work is timely, methodologically rigorous, and addresses a critical knowledge gap in arid land biogeochemistry. However, some aspects require clarification to strengthen the conclusions.
Response 1: Thank you for your positive responses and many helpful suggestions and recommendations on our manuscript. We carefully revised the manuscript according your comments. Please see our detailed revision on the following specific explains and on the revised manuscript.
Specific Comments
Comments 2: Material Transparency: The OTC material (typically transparent polycarbonate or glass) is not specified, which may affect light transmittance and humidity. Revision suggestion: Add OTC photos or schematic diagrams in the supplementary materials (e.g., Supplementary Figure S1).
Response 2: Thank you for pointing this out. We agree with this comment. Acrylic plate which is with high transparency, wear resistance and strong weather resistance, was used to set up the OTC. And we agree with your suggestion, although we have added the realistic picture of our study site, we further added one clearer picture of the OTC in Figure S1.
Comments 3: Spatial Representativeness: Is a 2 m × 2 m OTC sufficient to represent patchy desert vegetation? The authors need to justify whether this size adequately covers the spatial heterogeneity of soil microbes.
Response 3: Thank you for pointing out. When we established the experiment, we have also noticed this issue. We set 2 m × 2 m OTC based on two reasons: (1) 1 m × 1 m is enough to represent the desert vegetation based on our previous studies; (2) considering the spatial heterogeneity of soil microbes, the observation plots should be bigger than 1 m × 1 m; while, with the increasing size of OTC, the warming effects decrease, especially for the central area of the OTC. Thus, we set 2 m × 2 m OTC, it not only can represent the soil microbes, but also achieve warming effects. And we also added more words in the main text to sate this information (Lines 561-562).
Comments 4: Warming Method: Is the bottom of the warming device open? To my knowledge, OTCs usually cause soil cooling (negative warming), but this paper reports positive warming—why is that?
Response 4: Thank you very much for your careful comments on this detailed information. The bottom of the OTC is 3 cm higher than soil surface to ensure wind across. And based on our air temperature monitoring (Figure 1), OTC caused an average increase by 2.32 ℃. And we added one realistic picture of OTC in Figure S1.
Comments 5: Abbreviation Usage: "TBI" only appears in the Methods section but should be spelled out in full upon its first mention in the text.
Response 5: Thank you for your careful comments, we revised accordingly (Line 609).
Comments 6: Color Scheme in Figures: The shades of sky blue and light blue in the figures are difficult to distinguish.
Response 6: Thank you very much for your carful comments on the figure colors. We agree with this point and revised accordingly. Now, we used black, green, and blue colors in Figure 1, 3, and 4. The figures are clearer to see and the information is easier for readers.
Comments 7: Figure 2 Heatmap: The heatmap could convey more information. Typically, heatmap values can range from -1 to +1 to reflect positive/negative effects, with color intensity indicating the magnitude of change. Asterisks could also be added to denote significance.
Response 7: Thanks a lot for this detailed comment. It is worth to note that: (1) this heatmap is not to show the relationships among different parameters; (2) the different colors represent the values of different soil parameters which were standardized to a common scale ranging from 0 to 1. Thus, the heatmap values did not reflect positive/negative effects. And, it is not suitable to add asterisks to denote the significance, ANOVA results were presented in Table S1, and original data were shown in Table S2.
Reviewer 2 Report
Comments and Suggestions for Authors
Comments
- Are there no interval scales on the horizontal axis in Figures 1a and 1b?
- What important content does Figure 3 aim to convey? Are concise graphs or tables not sufficient to express it?
- How should the test-prepared fallen bags be properly placed at a depth of 3 cm?
- In Figure 5, how is the stabilization factor (s) on the horizontal axis defined? And how is it obtained?
- The data in Figure 5 is obtained from the literature, not from the data of this study. It is appropriate to present it in the discussion section.
- Which two directions does the bidirectional adjustment refer to? What is bidirectional adjustment?
- In Figure 7, the structural equation model has certain requirements for samples. Does the samples of each indicator in this study meet the requirements of the model?
- In the split plot experiment design, what is the difference between CK+CK and CK among the six treatments? Can CK+CK be treated as a single treatment?
- “In total, 180 Erodium oxyrhinchum litter bags, green tea bags, and red tea bags (6 treatments × 2 positions × 3 parallels × 5 replicates) were placed on 9 January 2023. All litter bags were harvested once a month, and the same number of three litter bags were re-placed. On 9 May 2023, we finished the final harvest, and there were a total of 2160 litter bags with the three litter types and four collective points.”
How were 2160 litter bags calculated?
- “We collected Erodium oxyrhinchum in the same experimental area, 2 g of Erodium oxyrhinchum were weighed and placed into 6.5 × 6.5 cm nylon bags which was similar with the tea bags” “3 cm belowground was placed to focus on the soil microbial effects”
Can the 6.5 × 6.5 cm nylon bags completely cover an area 3 cm below the ground?
- “After retrieval, each litter bag was cleaned to remove soils and roots, and dried again at 70°C until constant weight, mass loss at each time interval was determined.”
How to clean? How can it be considered as cleaned thoroughly?

Author Response
Thank you for your positive responses and many helpful suggestions and recommendations on our manuscript. We carefully revised the manuscript according your comments. Please see our detailed revision on the following specific explains and on the revised manuscript.
Comments 1: Are there no interval scales on the horizontal axis in Figures 1a and 1b?
Response 1: Thanks for your detailed comments, we agree this point, and we added the horizontal axis in Figure 1a and 1b.
Comments 2: What important content does Figure 3 aim to convey? Are concise graphs or tables not sufficient to express it?
Response 2: Thanks for your detailed comments. The “radar figure” can clearly present two information: (1) it can clearly show the differences among different soil enzymes; (2) it can clearly show which soil enzymes were highly impacted by treatments. The radar figure can clearly show more information than column graph or tables. Thus, we selected this figure type to show more information directly.
Comments 3: How should the test-prepared fallen bags be properly placed at a depth of 3 cm?
Response 3: Thank you very much for pointing out. We put the litter bags at soil surface and 3 cm below soil surface to reveal the potential influences of soil microbes on litter decomposition. The litter bags of green and red tea were vertically inserted into the soils, and litter bags of dominant species were inserted into the soils with an angle to ensure within 3 cm soil layer because that the litter bags of dominant species were 6.5 cm × 4.5 cm. We also added this information in the main text (Lines 588-590).
Comments 4: In Figure 5, how is the stabilization factor (s) on the horizontal axis defined? And how is it obtained?
Response 4: Thank you for pointing out, we agree with this point. The stabilization factor (S) represent the proportion of unstable components in the litter eventually stabilize and become recalcitrant. This calculation method was based on the “Tea Bag Index” research groups. We added the calculated equation and more information in the manuscript (Lines 607-610). Now, it would be clearer for readers.
Comments 5: The data in Figure 5 is obtained from the literature, not from the data of this study. It is appropriate to present it in the discussion section.
Response 5: Thank you for pointing out. We agree with this point. The data in Figure 5 was obtained from published literature. While, here we wanted to show the differences of k and S across different ecosystems, it is needed to present the results in the “Result” section. Thus, we preferred to put the Figure 5 in the results, not discussion section.
Comments 6: Which two directions does the bidirectional adjustment refer to? What is bidirectional adjustment?
Response 6: Thank you for pointing out. The bidirectional adjustment refers to two aspects: (1) winter snow changes and warming influenced litter decomposition through different pathways; winter snow changes indirectly impacted soil enzyme activities and litter decomposition through altering soil physical and chemical properties; while warming posed direct effects on soil enzyme activities and litter decomposition. (2) There were significantly interactive and superimposed effects of winter snow changes and warming on litter decomposition. Warming accelerated snow melting in early spring, while warming would aggravate soil moisture deficit during non-snow cover period, especially for the snow removal treatment. Considering the above two aspects, we think “bidirectional adjustment” can better represent the influence patterns compared to “interactive effects” or “synergistic effects”.
Comments 7: In Figure 7, the structural equation model has certain requirements for samples. Does the samples of each indicator in this study meet the requirements of the model?
Response 7: Thank you for pointing this analysis method out. We agree that structural equation model needs certain requirements of samples and moderate potential influence pathways in the model. For our SEM model, the SEM parameters, such as, p = 0.375 and 0.406, both higher than 0.05, indicating that SEM fits well with our dataset. Furthermore, Comparative Fit Index (CFI) is used to assess the fitness of SEM, CFI > 0.9 indicates the good fitness, the value is closer to 1, the better of the fitness. In our model, CFI was 0.995 and 0.996, respectively, further indicating that the SEM fitted well with our dataset.
Comments 8: In the split plot experiment design, what is the difference between CK+CK and CK among the six treatments? Can CK+CK be treated as a single treatment?
Response 8: Thank you for pointing out. We agree and revised it using “CK”. Now it is clearer to see for readers and can not cause misleading.
Comments 9: “In total, 180 Erodium oxyrhinchum litter bags, green tea bags, and red tea bags (6 treatments × 2 positions × 3 parallels × 5 replicates) were placed on 9 January 2023. All litter bags were harvested once a month, and the same number of three litter bags were re-placed. On 9 May 2023, we finished the final harvest, and there were a total of 2160 litter bags with the three litter types and four collective points.”
How were 2160 litter bags calculated?
Response 9: Thank you for pointing out. As we mentioned in the manuscript, there were 6 treatments, 2 positions, 3 parallels, 3 litter bag types, 4 sample times and 5 replicates, thus 2160 litter bags in total during our experimental period.
Comments 10: “We collected Erodium oxyrhinchum in the same experimental area, 2 g of Erodium oxyrhinchum were weighed and placed into 6.5 × 6.5 cm nylon bags which was similar with the tea bags” “3 cm belowground was placed to focus on the soil microbial effects”
Can the 6.5 × 6.5 cm nylon bags completely cover an area 3 cm below the ground?
Response 10: Thank you for pointing out. As mentioned above, this litter bag was inserted at an angle into the soil to ensure within 3 cm soil layer. We also added this information in the manuscript (Lines 588-590).
Comments 11: “After retrieval, each litter bag was cleaned to remove soils and roots, and dried again at 70°C until constant weight, mass loss at each time interval was determined.” How to clean? How can it be considered as cleaned thoroughly?
Response 11: Thank you for pointing out. The soil in our study site is sandy, thus the litter bag was easy to remove soils by slight shaking. For the roots and sandy soils are difficult to remove, we carefully cleaned it by water above 0.15 mm sieve mesh.
Reviewer 3 Report
Comments and Suggestions for Authors
1. How do the authors make sure that the Open Top Chambers (OTCs) used for warming simulations don't change things like humidity, wind, or light intensity in a way that makes the results hard to understand?
2. Since the study only lasted five months, do the authors think the results are a good representation of how decomposition works over time? Are there any plans to keep an eye on things over several seasons?
3. Can the authors explain and back up their claim that higher β-glucosidase activity might slow down the breakdown of litter by causing intermediate products to build up?
4. Did you check to see if the assumptions of normality and homogeneity of variance were true before using ANOVA and SEM? What did you do with data that wasn't normal?
5. The mesh size of the litter bags kept out macrofauna, but how did the authors take into account the possible activity of mesofauna (like mites and springtails) that could affect how quickly things break down?
6. The study talks about a possible bad home-field advantage. Could the authors explain how the makeup of the community or the chemistry of the litter may have affected this result?
7. Can the authors explain if the legacy effects of snow removal are the same in all months or mostly limited to April, as they seem to be?
8. How did the TBI-based global comparison take into account site-specific factors like soil texture, microbial community, or UV exposure when figuring out decomposition rates (k and S)?
9. The authors say that different enzymes, like BG, ALP, and NAG, are what make snow and non-snow periods different. Are there any clues about changes in microbial communities that could explain this enzymatic control?
Comments on the Quality of English Language
There are a number of spelling and grammar mistakes in the manuscript. authors should make changes to the manuscript's language to make it clearer and easier to read before it is published?
Author Response
Thank you for your positive responses and many helpful suggestions and recommendations on our manuscript. We carefully revised the manuscript according your comments. Please see our detailed revision on the following specific explains and on the revised manuscript.
Comments 1: How do the authors make sure that the Open Top Chambers (OTCs) used for warming simulations don't change things like humidity, wind, or light intensity in a way that makes the results hard to understand?
Response 1: Thank you for pointing out. We agree with this point. As you mentioned, OTC potentially affect humidity, wind, and light intensity. In our experiment, the OTC was 3 cm higher than soil surface, by doing this, wind is limited affected. And acrylic plate with high transparency was used to set up the OTC, light intensity is limited affected. For humidity, it is closely related to air temperature, when warming caused by OTC, the humidity would decrease. Although there are disadvantages of OTC (e.g., changes in humidity), it is still the most convenient, economical and effective equipment for warming, especially in the field.
Comments 2: Since the study only lasted five months, do the authors think the results are a good representation of how decomposition works over time? Are there any plans to keep an eye on things over several seasons?
Response 2: Thank you for pointing out. We agree with this point. Based on the “tea bag method”, the decomposition rates of green and red tea become stable after one month, thus harvest once a month is suitable to assess the litter decomposition. While, as you mentioned, we did not test the litter decomposition for a complete year, we also notice this issue. We have discussed this point in the discussion (Lines 523-524).
Comments 3: Can the authors explain and back up their claim that higher β-glucosidase activity might slow down the breakdown of litter by causing intermediate products to build up?
Response 3: Thank you for pointing out. This point is important for our results and discussion. Soil β-glucosidase is a crucial enzyme in cellulose saccharification. While, soil β-glucosidase activities are closely related to the intermediate products (cellobiohydrolases) and final products (glucose). Thus, the accumulation of cellobiohydrolases and glucose could inhibit soil soil β-glucosidase activities. More detailed information was presented in the discussion section (Lines 397-404).
Comments 4: Did you check to see if the assumptions of normality and homogeneity of variance were true before using ANOVA and SEM? What did you do with data that wasn't normal?
Response 4: Thank you for pointing out. We agree with this point and did the normality and homogeneity test before ANOVA and SEM analysis. We also added this information in the “Statistical analysis” section (Lines 649-651).
Comments 5: The mesh size of the litter bags kept out macrofauna, but how did the authors take into account the possible activity of mesofauna (like mites and springtails) that could affect how quickly things break down?
Response 5: Thank you for pointing out. We agree with this point. As you mentioned some other soil mesofauna could impact the litter decomposition. Although we have discussed this point, it is not enough in the former version. We have added more discussion in the manuscript (Lines 514-520).
Comments 6: The study talks about a possible bad home-field advantage. Could the authors explain how the makeup of the community or the chemistry of the litter may have affected this result?
Response 6: Thank you very much for this constructive suggestion. We agree with this point. As we know “home-field advantage” is affected by lots of factors, such as litter types, plant community, climate conditions, and soil microbes. In our study, we found negative effects of “home-field advantage” because the litter types, plant community, and soil water availability. We added more discussion in the manuscript (Lines 504-508). Now it could be clearer for readers.
Comments 7: Can the authors explain if the legacy effects of snow removal are the same in all months or mostly limited to April, as they seem to be?
Response 7: Thank you for pointing out. Based on our results, it seems that the legacy effects of snow removal were more pronounced in April. Because soil water became limit and temperature became high in May. Whether the legacy effects can also be observed in other months, we are not sure due to the lack of observational data. Furthermore, we also discussed this shortcoming in the discussion, and we should conduct long-term observations in the future research (Lines 523-525).
Comments 8: How did the TBI-based global comparison take into account site-specific factors like soil texture, microbial community, or UV exposure when figuring out decomposition rates (k and S)?
Response 8: Thank you for pointing out. We agree with this point. Litter decomposition process is affected by multiaspects, as you mentioned “site-specific factors”. Thus, the “Tea Bag Index” method was used to eliminate these influences. The changes in k and S can represent the synthetic effects of these factors on litter decomposition.
Comments 9: The authors say that different enzymes, like BG, ALP, and NAG, are what make snow and non-snow periods different. Are there any clues about changes in microbial communities that could explain this enzymatic control?
Response 9: Thank you for pointing out. We agree with this point. Although soil enzyme activities are widely used to represent the changes in soil microbial functions, the changes in soil microbial community should also be tested. In our study, we did not test the changes in soil microbial community, thus, we added this information in the discussion (Lines 401-417).
Comments 10: There are a number of spelling and grammar mistakes in the manuscript. Authors should make changes to the manuscript's language to make it clearer and easier to read before it is published?
Response 10: Thank you very much for this detailed suggestion. We agree with this point. We carefully checked the whole manuscript, and revised the spelling and grammar mistakes (Lines 15, 19-22, 54, 199-202, 345, 357, 430). Thanks again for your careful review.
Round 2
Reviewer 3 Report
Comments and Suggestions for Authors
NA